# Maintaining Adversarial Robustness in Continuous Learning

## Abstract

Adversarial robustness is essential for security and reliability of machine learning systems. However, adversarial robustness enhanced by defense algorithms is easily erased as the neural network's weights update to learn new tasks. To address this vulnerability, it is essential to improve the capability of neural networks in terms of robust continual learning. Specially, we propose a novel gradient projection technique that effectively stabilizes sample gradients from previous data by orthogonally projecting back-propagation gradients onto a crucial subspace before using them for weight updates. This technique can maintaining robustness by collaborating with a class of defense algorithms through sample gradient smoothing. The experimental results on four benchmarks including Split-CIFAR100 and Split-miniImageNet, demonstrate that the superiority of the proposed approach in mitigating rapidly degradation of robustness during continual learning even when facing strong adversarial attacks.

## 1 Introduction

Continual learning and adversarial robustness are distinct and important research directions in artificial intelligence, each of which has witnessed significant advances. The former addresses a critical challenge known as catastrophic forgetting, where a neural network trained on a sequential of new tasks typically exhibits a dramatic drop in its performance on previously learned tasks if the model cannot revisit the previous data Farajtabar et al. (2020). The latter focuses on developing defenses against adversarial attacks that can deceive models into confidently misclassifying objects by adding subtle targeted perturbations to the input images often imperceptible to human observers Silva & Najafirad (2020).

However, the evolution of the neural network's adversarial robustness in context of continuous learning remains underexplored. In our experiments, we observe that adversarial robustness enhanced by well-designed defense algorithms on previous data is easily lost when the neural network updates its weights to accommodate new tasks, resulting in a phenomenon similar to catastrophic forgetting. This presents an intriguing challenge: how can we maintain the adversarial robustness during continuous learning? In other words, the objective of continuous learning expands to concurrently encompass classification performance and adversarial robustness.

In this paper, we present a solution by proposing a novel gradient projection technique called Double Gradient Projection (DGP), which inherently enables collaboration with a class of defense algorithms that enhance robustness through sample gradient smoothing. DGP is grounded on a theoretical hypothesis that a neural network's robustness can be maintained if the smoothness of sample gradients from previous data remain unchanged after weight updates. Specifically, when learning a new task, DGP projects the back-propagation gradients into the orthogonal direction to a crucial subspace before utilizing them for weight updates. This gradient subspace consists of two sets of base vectors derived from previous tasks, which are obtained by performing singular value decomposition on the layer-wise outputs of the neural network and the gradients of layer-wise outputs with respect to samples, respectively. Our contributions are summarized as follows:

1. We introduce the problem of robust continual learning in the scenario where data from previous tasks cannot be revisited.

2. We propose the Double Gradient Projection approach that stabilizes the sample gradients from previous tasks by orthogonally constraining the direction of weight updates. It can maintain robustness by collaborating with a class of defense algorithms that enhance robustness through sample gradient smoothing.

3. We validate the superiority of our approach on four image benchmarks. Furthermore, the experiment results indicate that without a tailored design, direct combination of existing continual learning and defense algorithms into the training procedure can be conflicting, resulting that the efficacy of the former is seriously weakened.

## 2 BACKGROUND

In this section, we introduce the preliminary concepts underlying our work, including sample gradient smoothing and gradient projection.

### 2.1 SAMPLE GRADIENT SMOOTHING

**Input gradient regularization** (IGR) Ross & Doshi-Velez (2018). The robustness of the neural network trained with IGR has been demonstrated across multiple attacks, architectures and datasets. IGR optimizes a neural network $f_{\mathbf{w}}$ by minimizing both the classification loss and the rate of change of that loss with respect to samples, formulated as:

$$\mathbf{w}^* = \underset{\mathbf{w}}{\operatorname{argmin}} H\left(\mathbf{y}, \hat{\mathbf{y}}\right) + \lambda \left\|\nabla_{\mathbf{x}} H\left(\mathbf{y}, \hat{\mathbf{y}}\right)\right\|, \quad (1)$$

where $H\left(\cdot, \cdot\right)$ is the cross-entropy and $\lambda$ is a hyperparameter controlling the regular strength. The second term on the right side is to make the variation of the KL divergence between the final output $\hat{\mathbf{y}}$ and the label $\mathbf{y}$ become as small as possible if any sample $\mathbf{x}$ changes locally.

**Adversarial training (AT)** Goodfellow et al. (2016). AT enhances robustness by incorporating adversarial examples generated by Fast Gradient Sign Method (FGSM) Kurakin et al. (2018) into training data. Compared to IGR, which explicitly smooths sample gradients by adding a regularization term into the loss function, AT achieves gradient smoothing implicitly.

### 2.2 GRADIENT PROJECTION

Consider a sequence of task $\{\mathcal{T}_1, \mathcal{T}_2, \dots\}$ where task $\mathcal{T}_t$ is associated with paired dataset $\{\mathbf{X}_t, \mathbf{Y}_t\}$ of size $n_t$. When feeding data $\mathbf{X}_p$ from previous task $\mathcal{T}_p \left(p < t\right)$ into the neural network with optimal weight $\mathbf{W}_t$ for task $\mathcal{T}_t$ (see Fig. 1), the input

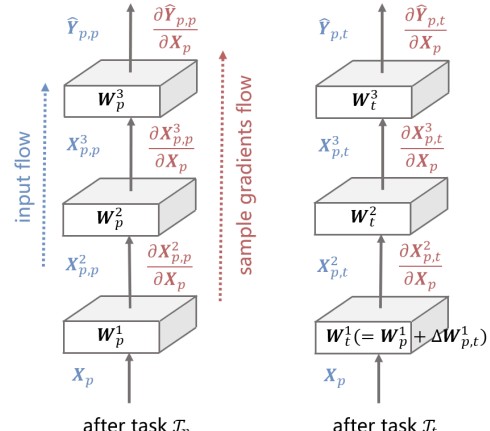

Figure 1: (Blue) Input flow. Feeding data $\mathbf{X}_p$ into an exemplar neural network after learning task $\mathcal{T}_p$ and $\mathcal{T}_t \left(p < t\right)$ respectively. $\Delta\mathbf{W}_{p,t}^l$ denotes the change of weights in task $\mathcal{T}_t$ relative to task $\mathcal{T}_p$. If $\Delta\mathbf{W}_{p,t}^1$ meets the constraint $\mathbf{X}_p\Delta\mathbf{W}_{p,t}^1 = 0$, then $\mathbf{X}_{p,p}^2$ is equal to $\mathbf{X}_{p,t}^2$. Recursively, the final outputs $\hat{\mathbf{Y}}_{p,t}$ and $\hat{\mathbf{Y}}_{p,p}$ will be identical even the weights of the neural network update; (Red) Sample gradients flow. If $\Delta\mathbf{W}_{p,t}^l$ meets the constraint $\frac{\partial\mathbf{X}_{p,t}^l}{\partial\mathbf{X}_p}\Delta\mathbf{W}_{p,t}^l = 0$, the sample gradients $\frac{\partial\hat{\mathbf{Y}}_{p,t}}{\partial\mathbf{X}_p}$ and $\frac{\partial\hat{\mathbf{Y}}_{p,p}}{\partial\mathbf{X}_p}$ will be identical.

and output of the $l$-th linear block (consisting of a linear layer and an activation function $\eta$) are denoted as $\mathbf{X}_{p,t}^l$ and $\mathbf{X}_{p,t}^{l+1}$ respectively, then

$$\mathbf{X}_{p,t}^{l+1} = \mathbf{X}_{p,t}^l \mathbf{W}_t^l \circ \eta = \mathbf{X}_{p,t}^l \left(\mathbf{W}_p^l + \Delta\mathbf{W}_{p,t}^l\right) \circ \eta, \quad (2)$$

where $\Delta\mathbf{W}_{p,t}^l$ denotes the change of weights in task $\mathcal{T}_t$ relative to task $\mathcal{T}_p$. Assuming $\mathbf{X}_{p,t}^l = \mathbf{X}_{p,p}^l$, a sufficient condition to guarantee $\mathbf{X}_{p,t}^{l+1} = \mathbf{X}_{p,p}^{l+1}$ is by imposing a constraint on $\Delta\mathbf{W}_{p,t}^l$ as Saha et al. (2021); Wang et al. (2021)

$$\mathbf{X}_{p,t}^l \Delta\mathbf{W}_{p,t}^l = 0. \quad (3)$$

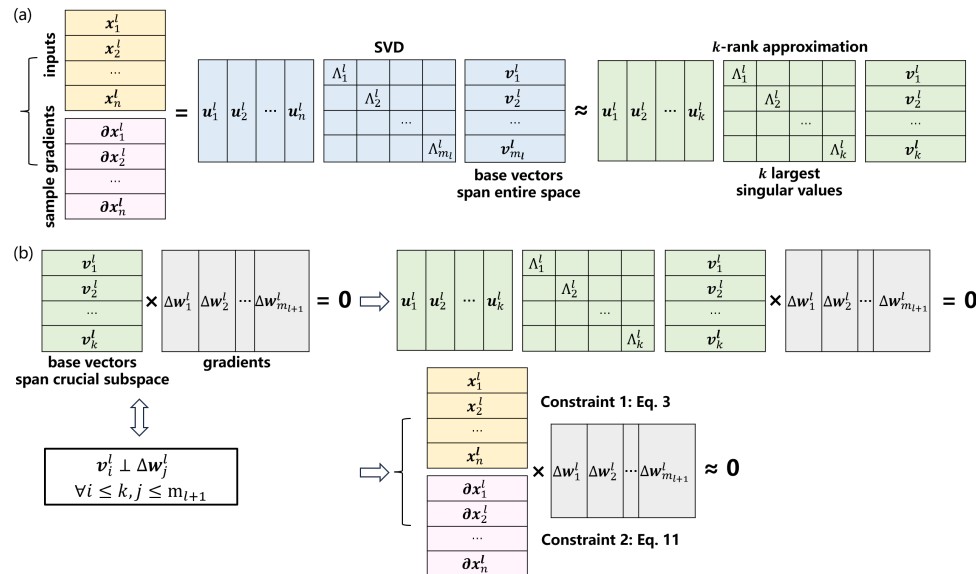

Figure 2: Graphical representation illustrating the imposed constraints in DGP. (a) The $\mathbf{X}^l$ or $\frac{d\mathbf{X}^l}{d\mathbf{X}}$ is approximated by $\mathbf{U}_k^l \mathbf{\Lambda}_k^l \left(\mathbf{V}_k^l\right)^{\mathrm{T}}$. (b) Multiplication of $\left(\mathbf{V}_k^l\right)^{\mathrm{T}}$ with $\Delta\mathbf{W}^l$ being zero implies that multiplication of $\mathbf{X}^l$ (or $\frac{\partial\mathbf{X}^l}{\partial\mathbf{X}}$) with $\Delta\mathbf{W}^l$ is approximately zero. Consequently, weight updates $\Delta\mathbf{W}^l$ have little impact on $\mathbf{X}^{l+1}$ (or $\frac{\partial\mathbf{X}^{l+1}}{\partial\mathbf{X}}$) of previous tasks.

The final output of a fully-connected network with $L$ linear blocks can be expressed as

$$\hat{\mathbf{Y}}_{p,t} = \mathbf{X}_p \mathbf{W}_t^1 \circ \eta \circ \mathbf{W}_t^2 \circ \cdots \circ \eta \circ \mathbf{W}_t^L, \tag{4}$$

where $\mathbf{X}_{p,t}^1 = \mathbf{X}_p$. If Eq. 3 is satisfied on each layer recursively, the final outputs $\hat{\mathbf{Y}}_{p,t}$ and $\hat{\mathbf{Y}}_{p,p}$ of the neural network with distinct weights for task $\mathcal{T}_t$ and task $\mathcal{T}_p$ are identical. Consequently, the performance on task $\mathcal{T}_p$ would be maintained after learning task $\mathcal{T}_t$.

Gradient Projection Memory (GPM) Saha et al. (2021), designed for improving continual learning ability, performs singular value decomposition (SVD) on $\mathbf{X}_{p,p}^l \in \mathbb{R}^{n \times m_l}$, where $n$ is the number of samples randomly drawn from the task $\mathcal{T}_p$ and $m_l$ is the number of features in an input of $l$-th layer:

$$\mathbf{X}_{p,p}^l \Delta\mathbf{W}_{p,t}^l = \mathbf{U}^l \mathbf{\Lambda}^l \left(\mathbf{V}^l\right)^{\mathrm{T}} \Delta\mathbf{W}_{p,t}^l \approx \mathbf{U}_k^l \mathbf{\Lambda}_k^l \left(\mathbf{V}_k^l\right)^{\mathrm{T}} \Delta\mathbf{W}_{p,t}^l, \tag{5}$$

where $\left(\mathbf{V}^l\right)^T \in \mathbb{R}^{m_l \times m_l}$ is an orthogonal matrix, of which all the row vectors as a basis span the entire $m_l$-dimensional space. Eq. 3 holds true when $\left(\mathbf{V}^l\right)^T \Delta\mathbf{W}_{p,t}^l = 0$, indicating that each column vector of $\Delta\mathbf{W}_{p,t}^l \in \mathbb{R}^{m_l \times m_{l+1}}$ is orthogonal to all the row vectors of $\left(\mathbf{V}^l\right)^T$. However, it is not possible for a $m_l$-dimensional vector to be orthogonal to the entire $m_l$-dimensional space unless it is the zero vector, implying no weight update. GPM approximates $\mathbf{X}_{p,p}^l$ as $\mathbf{U}_k^l \mathbf{\Lambda}_k^l \left(\mathbf{V}_k^l\right)^{\mathrm{T}}$, where $\left(\mathbf{V}_k^l\right)^{\mathrm{T}}$ preserves the first $k$ column vectors of $\left(\mathbf{V}^l\right)^{\mathrm{T}}$, corresponding to the $k$ largest singular values in diagonal matrix $\mathbf{\Lambda}^l$, and spans a subspace of $k \, (< m_l)$ dimensions. Among all subspaces of $k$ dimensions, weight update orthogonal to this crucial subspace allows for the maximal satisfaction of Eq. 3. An intuitive description is provided in Fig. 2. The value of $k$ is decided by the following criteria:

$$\left\|\mathbf{U}_k^l \mathbf{\Lambda}_k^l \left(\mathbf{V}_k^l\right)^{\mathrm{T}}\right\|_F^2 \geq \alpha^l \left\|\mathbf{U}^l \mathbf{\Lambda}^l \left(\mathbf{V}^l\right)^{\mathrm{T}}\right\|_F^2, \tag{6}$$

where $\alpha^l$ is a given threshold representing the trade-off between learning plasticity and memory stability of the neural network Wang et al. (2024). By establishing a dedicated pool $\mathcal{P}^l$ to retain base vectors $\left(\mathbf{V}_k^l\right)^{\mathrm{T}}$ from previous tasks, GPM enforces the orthogonality of gradients with respect to

these base vectors in the learning process of a new task:

$$\nabla_{W^l}\mathcal{L} = \nabla_{W^l}\mathcal{L} - (\nabla_{W^l}\mathcal{L})\,\mathcal{P}^l\left(\mathcal{P}^l\right)^T. \tag{7}$$

For the convolutional layer, the convolution operator can also be formulated as matrix multiplication. Please refer to Liu et al. (2018); Saha et al. (2021) for details.

## 3  METHOD

In this section, we propose a novel gradient projection technique inspired from GPM to tackle the challenge of maintaining adversarial robustness in a continuous learning scenario, where revisiting previous data is not feasible. We hypothesize that if we can stabilize the sample gradients smoothed by defense algorithms such as IGR and AT on previous tasks, the adversarial robustness of the neural network will hold even after its weights update for learning a sequence of new tasks.

### 3.1  CONSTRAINT ON WEIGHT UPDATES

By applying the chain rule for derivatives of composite functions, the gradient of the neural network's (with $L$ blocks) final output $\hat{\mathbf{y}}$ with respect to a sample $\mathbf{x}$ can be expressed in terms of recursive multiplication:

$$\frac{\partial \mathbf{x}^{(2)}}{\partial \mathbf{x}}\frac{\partial \mathbf{x}^{(3)}}{\partial \mathbf{x}^{(2)}}\cdots\frac{\partial \hat{\mathbf{y}}}{\partial \mathbf{x}^L} = \frac{\partial \hat{\mathbf{y}}}{\partial \mathbf{x}}. \tag{8}$$

We reformulate Eq. 8 in the Jacobian matrix form

$$\begin{bmatrix} \frac{\partial x_1^{(2)}}{\partial x_1} & \cdots & \frac{\partial x_{m_2}^{(2)}}{\partial x_1} \\ \vdots & \vdots & \vdots \\ \frac{\partial x_1^{(2)}}{\partial x_{m_1}} & \cdots & \frac{\partial x_{m_2}^{(2)}}{\partial x_{m_1}} \end{bmatrix}\begin{bmatrix} \frac{\partial x_1^{(3)}}{\partial x_1^{(2)}} & \cdots & \frac{\partial x_{m_3}^{(3)}}{\partial x_1^{(2)}} \\ \vdots & \vdots & \vdots \\ \frac{\partial x_1^{(3)}}{\partial x_{m_2}^{(2)}} & \cdots & \frac{\partial x_{m_3}^{(3)}}{\partial x_{m_2}^{(2)}} \end{bmatrix}\cdots\begin{bmatrix} \frac{\partial \hat{y}_1}{\partial x_1^L} & \cdots & \frac{\partial \hat{y}_c}{\partial x_1^L} \\ \vdots & \vdots & \vdots \\ \frac{\partial \hat{y}_1}{\partial x_{m_L}^L} & \cdots & \frac{\partial \hat{y}_c}{\partial x_{m_L}^L} \end{bmatrix} = \begin{bmatrix} \frac{\partial \hat{y}_1}{\partial x_1} & \cdots & \frac{\partial \hat{y}_c}{\partial x_1} \\ \vdots & \vdots & \vdots \\ \frac{\partial \hat{y}_1}{\partial x_{m_1}} & \cdots & \frac{\partial \hat{y}_c}{\partial x_{m_1}} \end{bmatrix},$$

$$\tag{9}$$

where $m_l$ represents the number of features in the input of $l$-th block, and $c$ equals the total number of classes within labels.

#### 3.1.1  LINEAR BLOCK

**Stringent guarantee**. The gradient of the output $\mathbf{x}^{l+1}$ with respect to the input $\mathbf{x}^l$ of the $l$-th block is derived as explicitly related to the weights $\mathbf{W}^l$:

$$\frac{\partial \mathbf{x}^{l+1}}{\partial \mathbf{x}^l} = \mathbf{W}^l\left(\eta'|\mathbf{x}^{l+1}\right) = \begin{bmatrix} w_{1,1}^l & \cdots & w_{m_{l+1},1}^l \\ \vdots & \vdots & \vdots \\ w_{1,m_l}^l & \cdots & w_{m_{l+1},m_l}^l \end{bmatrix}\begin{bmatrix} \eta'|x_1^{l+1} & \cdots & 0 \\ \vdots & \vdots & \vdots \\ 0 & \cdots & \eta'|x_{m_{l+1}}^{l+1} \end{bmatrix}. \tag{10}$$

Each column of the weight matrix (left) represents a single artificial neuron in the linear layer. Element $\eta'|x_i^{l+1}$ in the diagonal matrix (right) represents the derivative of activation function $\eta$ e.g., Relu, of which $\eta' = 1$ if activation $x_i^{l+1} > 0$, otherwise it is 0. By combining Eq. 8 and Eq. 10, we can efficiently compute the gradient of each block's input with respect to the sample based on that of the previous block, i.e., $\frac{\partial \mathbf{x}^{l+1}}{\partial \mathbf{x}} = \frac{\partial \mathbf{x}^l}{\partial \mathbf{x}}\frac{\partial \mathbf{x}^{l+1}}{\partial \mathbf{x}^l}$, instead of having to compute them from scratch which is time-consuming.

We then impose a constraint on weight updates for stabilizing sample gradients (the core idea of this work):

$$\frac{\partial \mathbf{X}_{p,t}^l}{\partial \mathbf{X}_p}\Delta \mathbf{W}_{p,t}^l = 0. \tag{11}$$

If Eq. 11 is satisfied on each layer recursively, the sample gradients $\frac{\partial \hat{\mathbf{Y}}_{p,t}}{\partial \mathbf{X}_p}$ and $\frac{\partial \hat{\mathbf{Y}}_{p,p}}{\partial \mathbf{X}_p}$ of a neural network with distinct weights for task $\mathcal{T}_t$ and task $\mathcal{T}_p$ are identical (see Fig. 1). Similarly, the method in GPM can be used for an approximate implementation of Eq.11.

**Weak guarantee**. However, directly performing SVD on the matrix $\frac{\partial \mathbf{X}^l}{\partial \mathbf{X}} \in \mathbb{R}^{(nm_1)\times m_l}$ is computationally time-consuming due to its large size, which is a concat of multiple $\frac{\partial \mathbf{x}^l}{\partial \mathbf{x}} \in \mathbb{R}^{m_1 \times m_l}$. To compress the matrix, we modify $\frac{\partial \mathbf{x}^{(2)}}{\partial \mathbf{x}}$ through column-wise summation, which is located at the beginning of the matrix chain as depicted in Eq. 8, and substitute it back into Eq. 9 as:

$$
\begin{bmatrix} \sum_{i=1}^{m_1} \frac{\partial x_1^{(2)}}{\partial x_i} & \cdots & \sum_{i=1}^{m_1} \frac{\partial x_{m_2}^{(2)}}{\partial x_i} \end{bmatrix}
\begin{bmatrix} \frac{\partial x_1^{(3)}}{\partial x_1^{(2)}} & \cdots & \frac{\partial x_{m_3}^{(3)}}{\partial x_1^{(2)}} \\ \vdots & \vdots & \vdots \\ \frac{\partial x_1^{(3)}}{\partial x_{m_2}^{(2)}} & \cdots & \frac{\partial x_{m_3}^{(3)}}{\partial x_{m_2}^{(2)}} \end{bmatrix}
\cdots
\begin{bmatrix} \frac{\partial \hat{y}_1}{\partial x_1^L} & \cdots & \frac{\partial \hat{y}_c}{\partial x_1^L} \\ \vdots & \vdots & \vdots \\ \frac{\partial \hat{y}_1}{\partial x_{m_L}^L} & \cdots & \frac{\partial \hat{y}_c}{\partial x_{m_L}^L} \end{bmatrix}
= \begin{bmatrix} \sum_{i=1}^{m_1} \frac{\partial \hat{y}_1}{\partial x_i} & \cdots & \sum_{i=1}^{m_1} \frac{\partial \hat{y}_c}{\partial x_i} \end{bmatrix}.
$$
(12)

According to Eq. 12, $\frac{\partial \mathbf{x}^l}{\partial \mathbf{x}}$ transforms to a vector within the space $\mathbb{R}^{m_l}$. This modification significantly reduces the computational time required for performing SVD on matrix $\frac{\partial \mathbf{X}^l}{\partial \mathbf{X}} \in \mathbb{R}^{n \times m_l}$, while relaxes the stringent guarantee for stabilizing $\frac{\partial \hat{\mathbf{y}}}{\partial \mathbf{x}}$ to a less restrictive one (see right-hand side of Eq. 9 and Eq. 12). The target of the constraint in Eq. 11 is altered from stabilizing the gradient of each final output with respect to each feature in the sample, to stabilizing the sum of gradients of each final output with respect to all features in the sample. This weak guarantee is sufficient to yield desirable results in our experiments for both fully-connected and convolutional neural networks.

### 3.1.2 CONVOLUTIONAL BLOCK

The convolutional block consists of a convolution layer, a batch normalization layer (BN) and an activated function. The gradient of the output $\mathbf{x}^{l+1}$ with respect to an input $\mathbf{x}^l$ is derived as:

$$
\frac{\partial \mathbf{x}^{l+1}}{\partial \mathbf{x}^l} = \widetilde{\mathbf{W}}^l \partial \mathrm{BN}^l \left( \eta' \mid \mathbf{x}^{l+1} \right),
$$
(13)

where $\partial \mathrm{BN}^l$ denotes the gradients in BN. The mean and variance $\sigma^2$ per-channel used for normalization are constants during evaluation, if they are calculated by tracking during training Ioffe & Szegedy (2015). In this case, $\partial \mathrm{BN}^l$ is a diagonal matrix with the diagonal element $\frac{\gamma}{\sqrt{\sigma^2 + \epsilon}}$. Please see Appendix A.4 for the case where the mean and variance are batch statistics.

There are two differences between the convolution layer and the linear layer. First, $\widetilde{\mathbf{W}}^l \in \mathbb{R}^{(c_l h_l \omega_l) \times (c_{l+1} h_{l+1} \omega_{l+1})}$ is distinct from the weight matrix $\mathbf{W}^l \in \mathbb{R}^{(c_l k_l k_l) \times c_{l+1}}$, where each column represents a flattened convolution kernel. Here, $c_{l+1}$ ($k_l$) denotes the number (size) of kernels in the $l$-th layer, and $h_l$ ($\omega_l$) denotes the height (width) of the input $\mathbf{x}^l$. We give a simple example to illustrate the composition of $\widetilde{\mathbf{W}}^l$ in Appendix Fig. 6. The $\widetilde{\mathbf{W}}^l$ is sparse, with non-zero elements only present at specific positions of each column, corresponding to the input features that interact with a convolution kernel. To circumvent the intricate construction of matrix $\widetilde{\mathbf{W}}^l$, we identify an alternative approach for implementing $\frac{\partial \mathbf{x}^{l+1}}{\partial \mathbf{x}^l}$: reshaping $\frac{\partial \mathbf{x}^l}{\partial \mathbf{x}}$ from $(c_l h_l \omega_l,)$ to $(c_l, h_l, \omega_l)$, feeding it into the $l$-th convolution layer, and subsequently reshaping the output from $(c_{l+1}, h_{l+1}, \omega_{l+1})$ back to $(c_{l+1} h_{l+1} \omega_{l+1},)$, i.e., $\frac{\partial \mathbf{x}^{l+1}}{\partial \mathbf{x}}$.

Second, the base vectors formed by performing SVD on $\frac{\partial \mathbf{X}^l}{\partial \mathbf{X}} \in \mathbb{R}^{n \times (c_l h_l \omega_l)}$ (computed through Eq. 12), can be used directly to constrain the updates $\Delta \widetilde{\mathbf{W}}^l$ rather than $\Delta \mathbf{W}^l$ (see Appendix Fig. 7 for details). Consequently, we perform SVD after reshaping $\frac{\partial \mathbf{X}^l}{\partial \mathbf{X}}$ into a matrix $\in \mathbb{R}^{(n h_{l+1} \omega_{l+1}) \times (c_l k_l k_l)}$. This results that the base vectors have the same shape $(c_l k_l k_l,)$ with the flattened convolution kernels in the $l$-th layer, and can be used directly to constrain their weight updates.

## 3.2 DOUBLE GRADIENT PROJECTION (DGP)

The fundamental principle of our algorithm is concise: stabilizing the smoothed sample gradients (some implementation details are elaborated in the preceding subsection). The overall algorithmic flow is outlined as follows: Firstly, the neural network is trained on task $\mathcal{T}_t$ with a class of defense algorithms through sample gradient smoothing. The weight update is projected to be orthogonal to

all the base vectors in pool $\mathcal{P}$ if the sequential number $t > 1$. Subsequently, after training, SVD is performed on the layer-wise outputs $\mathbf{X}_t^l$ to obtain base vectors for stabilizing the final outputs of the neural network on task $\mathcal{T}_t$. Lastly, another SVD is performed on the gradients of the layer-wise outputs with respect to the samples $\frac{\partial \mathbf{X}_t^l}{\partial \mathbf{X}_t}$ to obtain base vectors for stabilizing the gradients of final outputs with respect to the samples on task $\mathcal{T}_t$. Note that in order to eliminate the redundancy between new bases and existing bases in the pool $\mathcal{P}$, both $\mathbf{X}_t^l$ and $\frac{\partial \mathbf{X}_t^l}{\partial \mathbf{X}_t}$ are projected orthogonally onto $\mathcal{P}^l$ prior to performing SVD. A compact pseudo-code of our algorithm is presented in Alg. 1.

---

**Algorithm 1** Double Gradient Projection

---

**Input**: Training dataset $\{\mathbf{X}_t, \mathbf{Y}_t\}$ for task $\mathcal{T}_t \in \{\mathcal{T}_1, \mathcal{T}_2, \dots\}$, regularization strength $\lambda$ and learning rate $\alpha$

**Output**: Neural network $f_w$ with optimal weights

**Initialization**: Pool $\mathcal{P} \leftarrow \{\}$

1: **for** task $\mathcal{T}_t \in \{\mathcal{T}_1, \mathcal{T}_2, \dots\}$ **do**
2:    **while** not converged **do**
3:       Sample a batch from $\{\mathbf{X}_t, \mathbf{Y}_t\}$ and Calculate $\{\nabla_{\mathbf{W}^l}\mathcal{L}\}$
4:       **if** $t > 1$ **then**
5:          $\nabla_{W^l}\mathcal{L} = \nabla_{W^l}\mathcal{L} - (\nabla_{W^l}\mathcal{L}) \mathcal{P}^l (\mathcal{P}^l)^T \ \forall l = 1, 2, ..., L$   ▷ *Gradient projection for layers*
6:       **end if**
7:       $\mathbf{W}^l \leftarrow \mathbf{W}^l - \alpha \nabla_{\mathbf{W}^l}\mathcal{L}$                         ▷ *Weight updates*
8:    **end while**
9:    $\mathbf{X}_t^l \leftarrow \mathbf{X}_t^l - \mathcal{P}^l (\mathcal{P}^l)^T \mathbf{X}_t^l$            ▷ *Ensure uniqueness for new bases*
10:   Perform SVD on $\mathbf{X}_t^l$ and put bases into $\mathcal{P}^l$     ▷ *Construct the first set of bases*
11:    $\frac{\partial \mathbf{X}_t^l}{\partial \mathbf{X}_t} \leftarrow \frac{\partial \mathbf{X}_t^l}{\partial \mathbf{X}_t} - \mathcal{P}^l (\mathcal{P}^l)^T \frac{\partial \mathbf{X}_t^l}{\partial \mathbf{X}_t}$
12:   Perform SVD on $\frac{\partial \mathbf{X}_t^l}{\partial \mathbf{X}_t}$ and put bases into $\mathcal{P}^l$    ▷ *Construct the second set of bases*
13: **end for**

---

## 4 EXPERIMENT

### 4.1 STEP

**Baselines**. For continual learning Lomonaco et al. (2021), in addition to SGD, which serves as a naive baseline using stochastic gradient descent to optimize the neural network, we adopt six algorithms cover three most important techniques in the field of continual learning: regularization – EWC Kirkpatrick et al. (2017) and SI Zenke et al. (2017), memory replay – GEM Lopez-Paz & Ranzato (2017) and A-GEM Chaudhry et al. (2019a), and gradient projection – OGD Farajtabar et al. (2020) and GPM Saha et al. (2021). The fundamental principle of each algorithm are outlined in Appendix B.3. For adversarial robustness, we adopt IGR Ross & Doshi-Velez (2018) and AT Kurakin et al. (2018).

We combine algorithms from fields of continual learning and adversarial robustness, such as EWC + IGR, to establish the baselines for robust continual learning. On the other hand, we apply the FGSM Kurakin et al. (2018), PGD Madry et al. (2018), and AutoAttack Croce & Hein (2020) to generate adversarial samples.

**Metrics**. We use average accuracy (ACC) and backward transfer (BWT) defined as

$$\text{ACC} = \frac{1}{T} \sum_{t=1}^{T} R_{T,t}, \qquad\qquad \text{BWT} = \frac{1}{T-1} \sum_{t=1}^{T-1} R_{T,t} - R_{t,t}, \qquad (14)$$

where $R_{T,t}$ denotes the accuracy of task $t$ at the end of learning task $T$. To evaluate the performance of continuous learning, we measure accuracy on test data from previous tasks. To evaluate the adversarial robustness, we then perturb test data, and re-measure accuracy on the corresponding adversarial samples.

**Benchmarks**. We evaluate our approach on four supervised benchmarks. Permuted MNIST and Rotated MNIST are variants of MNIST dataset with 10 tasks applying random permutations of the

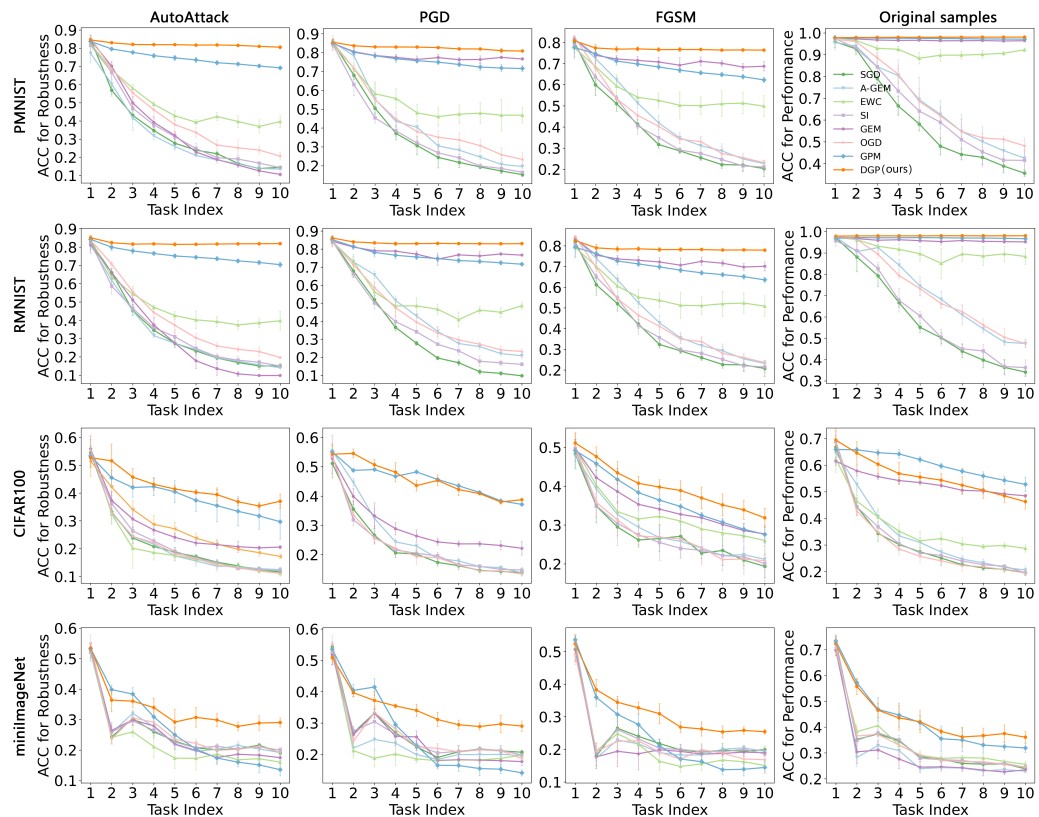

Figure 3: ACC varying with the number of learned tasks on datasets of Permuted MNIST (first row), Rotated MNIST (second row), CIFAR100 (third row) and miniImageNet (fourth row). ACC is measured on adversarial samples generated by AutoAttack (first column), PGD (second column) and FGSM (third column), as well as original samples (fourth column). The horizontal axis indicates the number of tasks learned by the neural network at present. The defense algorithm used here is IGR. Errors bars denote standard deviation.

input pixels and random rotations of the original images respectively Goodfellow et al. (2014); Liu & Liu (2022). Split-CIFAR100 Zenke et al. (2017) is a random division of CIFAR100 into 10 subsets, each with 10 different classes. Split-miniImageNet is a random division of a part of the original ImageNet dataset Chaudhry et al. (2020) into 10 subsets, each with 5 different classes. All images for a specific class are exclusively present in one subset and no overlap of classes between subsets, thus these subsets can be considered as independent datasets, representing a sequence of 10 classification tasks.

**Architectures**: The neural network architecture varies across experiments: a fully connected network is used for the MNIST experiments, an AlexNet for the Split-CIFAR100 experiment, and a variant of ResNet18 for the Split-miniImageNet experiment. In both Split-CIFAR100 and Split-miniImageNet experiments, each task has an independent classifier without constraints on weight updates.

The values of $\frac{\partial \mathbf{x}^2}{\partial \mathbf{x}}$ (initial term in Eq.10) are solely determined by the weights of the first layer when feeding the same samples (see Fig.1). There are two options to make $\frac{\partial \mathbf{x}_{p,t}^2}{\partial \mathbf{x}_p} = \frac{\partial \mathbf{x}_{p,p}^2}{\partial \mathbf{x}_p}$: fixing the first layer after learning task $p$ or assigning an independent first layer to each task. The latter option is chosen for our experiments, as the former seriously diminishes the neural network's learning ability in subsequent tasks. To ensure fairness, the same setup is applied to the baselines. Further details on architectures can be found in Appendix B.2.

**Training details**: For the MNIST experiments, the batch size, number of epochs, and input gradient regularization $\lambda$ are set to 32/10/50, respectively. For the Split-CIFAR100 experiments, the values are 10/100/1, and for the Split-miniImageNet experiments, they are 10/20/1. SGD is used as the optimizer. The hyperparameter configurations for adversarial attack and continuous learning algorithms are provided in Appendix B.3. All reported results are averaged over 5 runs with different seeds. We run the experiments on a local machine with three A800 GPUs.

## 4.2 RESULTS

### 4.2.1 ADVERSARIAL ROBUSTNESS

The results about robustness analysis on various datasets are presented in left three columns of Fig. 3, where different color lines represent the combinations by IGR with diverse continuous learning algorithms and DGP. Under attacks with increasing strengths (AutoAttack > PGD > FGSM), the proposed approach (orange lines) consistently exhibits a high level of effectiveness in maintaining robustness of neural networks enhanced by IGR. In contrast, the baseline such as IGR+GEM (purple lines), which performs well on MNIST datasets against PGD and FGSM attacks, demonstrates a significant decrease when fronted with AutoAttack. The advantage of our approach becomes even more evident when the number of learned tasks increases.

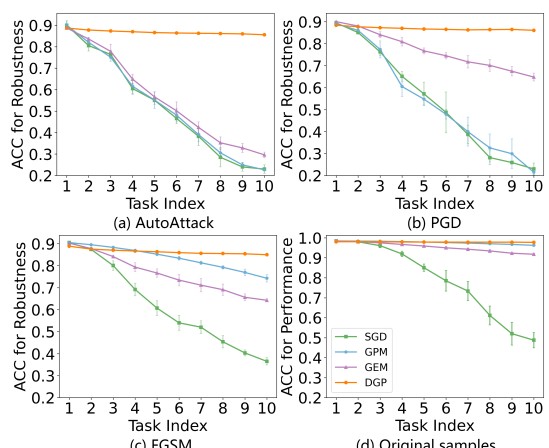

Figure 4: As Fig. 3, but for defense algorithm Adversarial Training (AT) on PMNIST dataset. Here, we combine AT with continual learning algorithms GEM and GPM, which have shown superior ACC compared to other baselines in Fig. 3.

The results of maintaining the robustness enhanced by AT are presented in Fig. 4. The results further demonstrates that baselines fail to effectively maintain the robustness enhanced by AT against AutoAttack and PGD attacks after the neural network learns a sequence of new tasks, whereas GDP performs well. Compared to Fig. 3, the advantage of the proposed method than baselines is more pronounced in Fig. 4.

Considering the collective insights presented in Figs. 3 and 4, it is crucial to underscore that the pursuit of an effective defense demands a tailored algorithm adept at accommodating variations in neural network's parameters. Direct combinations of existing defense strategies and continual learning methods, as demonstrated in our experiments, fall short of achieving the desired goal of continuous robustness.

### 4.2.2 CLASSIFICATION PERFORMANCE

We also assess the ability of the proposed approach for continual learning (ACC on original samples), as illustrated in the fourth column of Fig. 3. Our DGP algorithm demonstrates comparable performance to GPM and GEM on datasets of Permuted MNIST and Rotated MNIST, effectively addressing the issue of catastrophic forgetting on these two datasets. However, results on the datasets of Split-CIFAR indicate that the performance of DGP is slightly inferior to GPM. We speculate that the reason for this could be that DGP stores a larger number of bases after each task than GPM, as DGP constrains the weight updates to be orthogonal to two sets of base vectors – one for stabilizing the final output (required in GPM) and another for stabilizing the sample gradients. Orthogonality to more base vectors restricts weight updates to a narrower subspace, thereby limiting the plasticity of the neural network. Overall, our approach effectively maintains adversarial robustness while exhibiting continual learning ability.

In addition, it is noteworthy that the performance curves of many well-known continual learning algorithms (e.g., EWC) closely approximate that of naive SGD (green lines). This important observation suggest a potential incompatibility between existing defense (here IGR) and continuous learning algorithms. The effectiveness of the latter can be significantly weakened when they are mixed into the training process. For instance, both EWC and IGR add a regularization term into the loss function, but their guidance on the direction of weight updates interferes with each other. Additional experiment results of the incompatibility with another defense algorithm Distillation Papernot et al. (2016) are shown in Appendix B.4.

### 4.2.3 STABILIZATION OF SAMPLE GRADIENTS

Our approach maintains adversarial robustness by stabilizing the smoothness of sample gradients. To valid its stabilization effect, we record the variation of sample gradients on the first task during continuous learning process. Specifically, we randomly select $n$ samples at the end of learning $\mathcal{T}_1$ and compute their gradients related to correspondingly final outputs. After learning each new task $\mathcal{T}_t$, we recompute their gradients. The variation of gradients between $\mathcal{T}_1$ and $\mathcal{T}_t$ is quantified by similarity measure:

$$\text{Sim} = \frac{\mathbf{g}_1 \mathbf{g}_t}{|\mathbf{g}_1| \, |\mathbf{g}_t|}, \qquad (15)$$

where $\mathbf{g}$ is a flattened vector of sample gradients. The results on various datasets are presented in Fig. 5. The orange line (representing DGP) shows a relatively flat downward trend, demonstrating the proposed approach indeed has the effect of stabilizing the sample gradients of previous tasks, as the neural network's weights update.

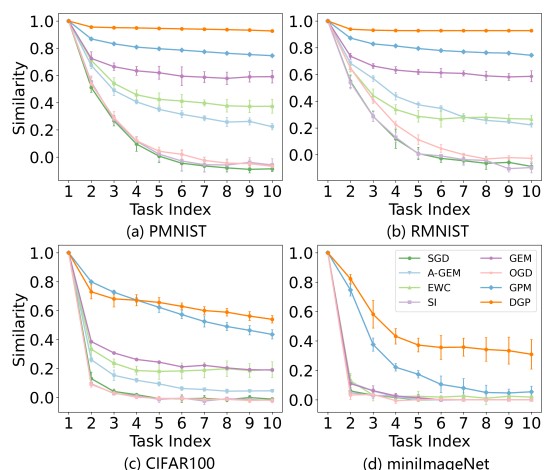

Figure 5: Gradient variation of samples from the first task $\mathcal{T}_1$ during continuous learning process trained with IGR. The variations are quantified through similarity.

## 5 RELATED WORKS

The aim of robust continual learning is not to
achieve stronger robustness on a single dataset, but rather to maintain robustness across multiple datasets encountered sequentially. One related work Bai et al. (2023) also explores the under-researched direction of robust continual learning. A fundamental distinction between that work and ours is their approach requires partial data from previous tasks to be accessible, thereby focusing on the selection of a key subset of previous data and optimizing its re-utilization in new tasks. In contrast, we follow the stricter yet realistic scenario in the field of continual learning that data from the previous tasks cannot be revisited. Further insights in the advancements in adversarial robustness and continual learning can be found in dedicated surveys Silva & Najafirad (2020); Wang et al. (2024).

## 6 LIMITATION AND DISCUSSION

In this work, we observe that the adversarial robustness gained by well-design defense algorithms is easily erased when the neural network learns new tasks. Direct combinations of existing defense and continuous learning algorithms fail to effectively address this issue, and may even give rise to conflicts between them. Therefore, we propose a novel gradient projection technique that can mitigate rapidly degradation of robustness in the face of drastic changes in model weights by collaborating with a class of defense algorithms through sample gradient smoothing.

According to our experiment, the proposed approach has certain limitations. First, as the number of base vectors becomes large, the stability of the neural network is enhanced to hold both robustness

and performance across previous tasks. This stability may restrict the plasticity of the neural network, potentially reducing its ability to learn new tasks. Second, if there are numerous tasks and the matrix consisting of orthogonal bases reaches full rank, we approximate this matrix by performing SVD and selecting column vectors corresponding to a fraction of the largest singular values as the new orthogonal bases to free up rank space. Three, due to the extra challenge posed by our problem, the perturbation size of adversarial attacks under which the proposed method work effectively is slightly smaller than typical values in adversarial robustness literature (Please see Appendix B.3 for more details).

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

# A    METHOD

## A.1    SAMPLE GRADIENTS

The sample gradients we stabilize in Sec. Method refer to the gradients of the final outputs with respect to samples, rather than gradients of the loss with respect to samples, which are penalized in IGR. Here, we show their relationship:

$$\frac{\partial \mathcal{L}}{\partial \mathbf{x}} = \frac{\partial \mathcal{L}}{\partial \hat{\mathbf{y}}} \frac{\partial \hat{\mathbf{y}}}{\partial \mathbf{x}} = g\left(\hat{\mathbf{y}}\right) \frac{\partial \hat{\mathbf{y}}}{\partial \mathbf{x}}, \tag{16}$$

Where $g$ is a function of $\hat{\mathbf{y}}$. The stabilization of both final outputs $\hat{y}$ and sample gradients $\frac{\partial \hat{\mathbf{y}}}{\partial \mathbf{x}}$ together can result in the stabilization of $\frac{\partial \mathcal{L}}{\partial \mathbf{x}}$. To maintain the adversarial robustness, achieved by reducing the sensitive of predictions (i.e., final outputs) to subtle changes in samples, it is sufficient to stabilize the smoothed $\frac{\partial \hat{\mathbf{y}}}{\partial \mathbf{x}}$.

## A.2    MATRIX COMPOSITION

A simple example to illustrate the composition of $\widetilde{\mathbf{W}}^l$ is depicted in Fig. 6.

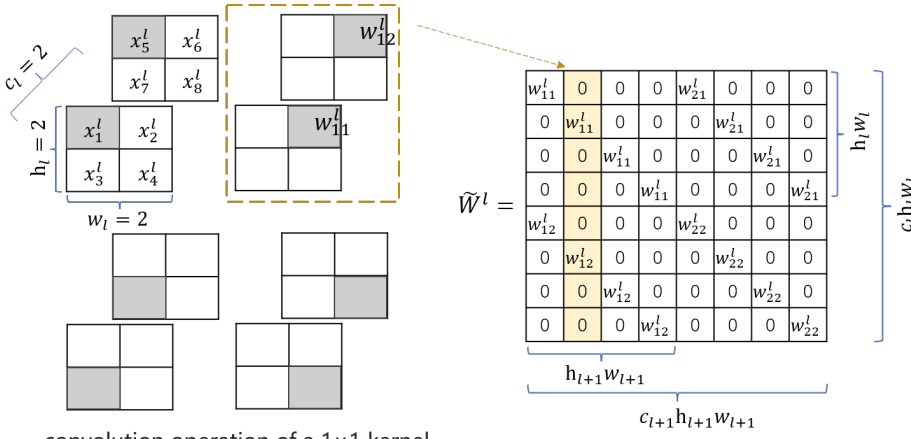

Figure 6: Graphic illustration of an example $\widetilde{\mathbf{W}}^l$. The shape of an example input $\mathbf{x}^l$ and a convolutional kernel $\mathbf{w}_i^l$ of $l$-th layer is $(2, 2, 2)$ and $(2, 1, 1)$ respectively. Suppose there are two convolution kernels in total, i.e., $c_{l+1} = 2$. The length of each column vector in $\widetilde{\mathbf{W}}^l$ is same as the flattened $\mathbf{x}^l$, i.e., $c_l h_l \omega_l = 2 \times 2 \times 2 = 8$. The four subplots in left display the convolution operation of the kernel $\mathbf{w}_1^l$ on $\mathbf{x}^l$, with grey checks indicating the specific input features on which the kernel acts after each slide. The four subplots sequentially correspond to the first four columns of the example $\widetilde{\mathbf{W}}^l$. The non-zero elements within each column of $\widetilde{\mathbf{W}}^l$ only present (filled by weights of a kernel) at positions corresponding to those specific input features, while the remains are zero-filled.

## A.3    RESHAPE $\frac{\partial \mathbf{X}^l}{\partial \mathbf{X}}$ PRIOR TO PERFORMING SVD

A simple example to illustrate why and how to reshape $\frac{\partial \mathbf{X}^l}{\partial \mathbf{X}}$ is depicted in Fig. 7.

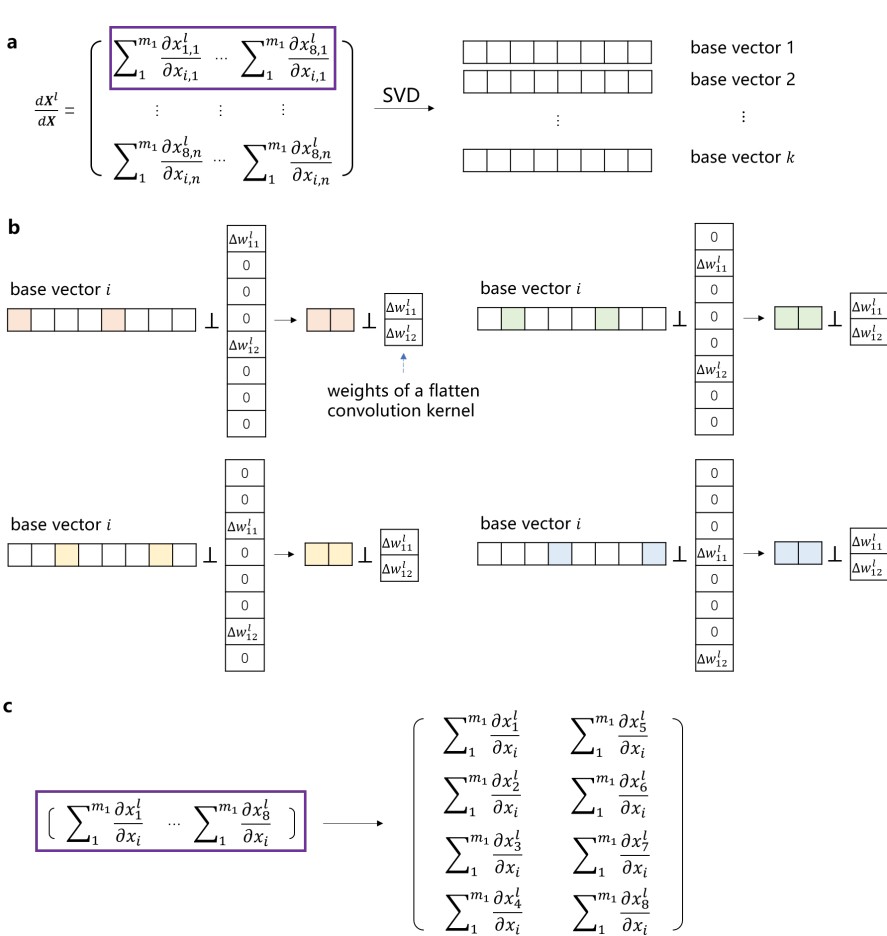

Figure 7: (a) Performing SVD on an example $\frac{\partial \mathbf{X}^l}{\partial \mathbf{X}}$ with the shape $(n, c_l h_l w_l)$ obtains the base vectors that can constrain the updates $\Delta \widetilde{\mathbf{W}}^l$. Here, $x_{i,j}^l$ denotes the $i$-th feature of $j$-th input of $l$-th layer, and an single input $\mathbf{x}^l$ from $\mathbf{X}^l$ is illustrated on the left of Fig. 6. (b) The orthogonality between any base vector and each $h_{l+1} w_{l+1} (= 4)$ column vectors of $\widetilde{\mathbf{W}}^l$ (see the right of Fig.6) is equivalent to the orthogonality between $h_{l+1} w_{l+1}$ sub-vectors of the base vector and the weight $\mathbf{w}^l$ of the kernel. (c) Prior to performing SVD, each row vector in $\frac{\partial \mathbf{X}^l}{\partial \mathbf{X}}$ is reshaped into a matrix consisting of $h_{l+1} w_{l+1}$ row vectors with a length of $c_l k_l k_l$. Consequently, the shape of $\frac{\partial \mathbf{X}^l}{\partial \mathbf{X}}$ is modified to $(n h_{l+1} w_{l+1}, c_l k_l k_l)$. This results that the base vectors obtained from performing SVD on $\frac{\partial \mathbf{X}^l}{\partial \mathbf{X}}$ have the same shape $(c_l k_l k_l, )$ with the flattened convolution kernel in the $l$-th layer, and can be directly used to constrain the weight updates of the convolution kernels.

## A.4 Gradients in Batch Normalization Layer

The batch normalization (BN) operation is formalized as

$$x_i^{out} = \frac{x_i^{in} - \mu}{\sqrt{\sigma^2 + \epsilon}} * \gamma + \beta, \tag{17}$$

Where $\gamma$ and $\beta$ is the learnable weights. When the mean $\mu$ and variance $\sigma^2$ per-channel are batch statistics, $x_i^{out}$ (feature $i$ in the output of BN) is not only correlated with $x_i^{in}$ (feature $i$ in the input of BN), but with the other features of the same channel in the whole batch samples. Therefore, the Jacobian matrix of the $l$-th BN (after $l$-th convolution layer, as shown in Eq. 13 of main text) is a matrix across batch samples with the shape $(nc_{l+1}h_{l+1}\omega_{l+1}, nc_{l+1}h_{l+1}\omega_{l+1})$, where each element $\frac{\partial x_i^{out}}{\partial x_j^{in}}$ is given by:

$$\begin{cases} \gamma \left[ -\left(1 - \frac{1}{n}\right) \frac{1}{\sqrt{\sigma^2+\epsilon}} - \frac{1}{(n-1)(\sigma^2+\epsilon)^{\frac{3}{2}}} \left(x_i - \mu\right)^2 \right] & \text{if } i = j, \\ \\ \gamma \left[ -\frac{1}{n\sqrt{\sigma^2+\epsilon}} - \frac{1}{(n-1)(\sigma^2+\epsilon)^{\frac{3}{2}}} \left(x_i - \mu\right)\left(x_j - \mu\right) \right] & \text{if } i \neq j \text{ and } x_i, x_j \text{ are in same channel,} \\ \\ 0 & \text{others.} \end{cases} \tag{18}$$

However, due to its extensive scale, the storage or computation of this Jacobian matrix poses high hardware requirements. To facilitate implementation, we propose decomposing this Jacobian matrix into $c_{l+1}$ submatrices with the shape $(nh_{l+1}w_{l+1}, nh_{l+1}w_{l+1})$, of which all elements belong to the same channel. Subsequently, these submatrices are concatenated to form a new matrix with shape $(c_{l+1}, nh_{l+1}w_{l+1}, nh_{l+1}w_{l+1})$, which effectively optimizes memory usage by eliminating a significant number of zero elements compared to the original Jacobian matrix. Before multiplying with this new matrix, the input gradient matrix of BN should be reshaped from $(n, c_{l+1}h_{l+1}w_{l+1})$ to $(c_{l+1}, nh_{l+1}w_{l+1})$.

## A.5 Computational Complexity Increment

Compared to the naive training procedure, i.e., SGD, the increase of training complexity in the proposed method is mainly related to SVD. After finishing the training on each new task, we gather the layer-wise outputs and their gradient with respect to samples, and perform SVD on them to obtain the base vectors used for gradient projection. Specifically, we call the interface in Pytorch to perform SVD decomposition. This interface uses the Jacobi method with a time complexity of approximately $o\left(nm_l \min\left(n, m_l\right)\right)$, where $n$ is the sample number and $m_l$ is the features number of the $l$-th layer's output. Assuming that the neural network consists of $L$ layers, with each layer's outputs having an equal number of features $m$, the proposed method introduces a computational complexity increment of $o\left(Lnm_l \min\left(n, m_l\right)\right)$.

# B Experiment

## B.1 ACC and BWT on various datasets

The comparisons of ACC and BWT, after learning all the tasks, are presented in Tab. 1 for Permuted MNIST, Tab. 2 for Rotated MNIST and Tab. 3 for Split-CIFAR100 datasets. The majority of baselines exhibit low accuracy (approaching random classification) quickly on the Split-miniImageNet dataset. Therefore, we do not compute BTW for Split-miniImageNet dataset.

| Method | Permuted MNIST | | | | | | | |
| | AutoAttack | | PGD | | FGSM | | Original samples | |
| | ACC(%) | BWT | ACC(%) | BWT | ACC(%) | BWT | ACC(%) | BWT |
|---|---|---|---|---|---|---|---|---|
| SGD | 14.1 | -0.75 | 15.4 | -0.74 | 21.8 | -0.67 | 36.8 | -0.66 |
| SI | 14.3 | -0.76 | 16.5 | -0.76 | 22.3 | -0.68 | 36.9 | -0.67 |
| A-GEM | 14.1 | -0.69 | 19.7 | -0.66 | 22.9 | -0.67 | 48.4 | -0.54 |
| EWC | 39.4 | -0.47 | 43.1 | -0.48 | 50.0 | -0.35 | 84.9 | -0.12 |
| GEM | 12.1 | -0.73 | 75.5 | -0.09 | 72.8 | -0.09 | 96.4 | **-0.01** |
| OGD | 19.7 | -0.72 | 24.1 | -0.67 | 26.0 | -0.63 | 46.8 | -0.57 |
| GPM | 70.4 | -0.11 | 72.9 | -0.10 | 65.7 | -0.12 | 97.2 | **-0.01** |
| DGP | **81.6** | **-0.01** | **81.2** | **-0.01** | **75.8** | **-0.03** | **97.6** | **-0.01** |

Table 1: Comparisons of ACC and BWT after learning all the tasks on the Permuted MNIST dataset.

| Method | Rotated MNIST | | | | | | | |
| | AutoAttack | | PGD | | FGSM | | Original samples | |
| | ACC(%) | BWT | ACC(%) | BWT | ACC(%) | BWT | ACC(%) | BWT |
|---|---|---|---|---|---|---|---|---|
| SGD | 14.1 | -0.76 | 9.9 | -0.76 | 20.4 | -0.69 | 32.3 | -0.71 |
| SI | 13.9 | -0.77 | 15.3 | -0.73 | 20.1 | -0.70 | 33.0 | -0.72 |
| A-GEM | 14.1 | -0.69 | 21.6 | -0.69 | 24.8 | -0.63 | 45.4 | -0.57 |
| EWC | 45.1 | -0.42 | 49.5 | -0.36 | 46.5 | -0.25 | 80.7 | -0.18 |
| GEM | 11.9 | -0.73 | 76.5 | -0.08 | 74.4 | -0.08 | 96.7 | -0.01 |
| OGD | 19.7 | -0.72 | 23.8 | -0.68 | 23.8 | -0.64 | 48.0 | -0.55 |
| GPM | 68.8 | -0.1 | 71.5 | -0.11 | 65.9 | -0.12 | 97.1 | -0.01 |
| DGP | **81.6** | **0.02** | **82.6** | **0.01** | **78.6** | **-0.01** | **98.1** | **-0.00** |

Table 2: Comparisons of ACC and BWT after learning all the tasks on the Rotated MNIST dataset.

| Method | Split-CIFAR100 | | | | | | | |
| | AutoAttack | | PGD | | FGSM | | Original samples | |
| | ACC(%) | BWT | ACC(%) | BWT | ACC(%) | BWT | ACC(%) | BWT |
|---|---|---|---|---|---|---|---|---|
| SGD | 10.3 | -0.45 | 12.8 | -0.45 | 46.5 | -0.25 | 19.4 | -0.49 |
| SI | 13.0 | -0.45 | 15.2 | -0.43 | 45.4 | -0.28 | 19.8 | -0.48 |
| A-GEM | 12.6 | -0.46 | 12.9 | -0.43 | 40.6 | -0.33 | 20.7 | -0.48 |
| EWC | 12.6 | -0.43 | 23.2 | -0.31 | 56.8 | -0.15 | 30.5 | -0.35 |
| GEM | 21.2 | -0.33 | 19.4 | -0.36 | 60.6 | -0.11 | 47.7 | -0.13 |
| OGD | 11.8 | -0.45 | 14.1 | -0.44 | 44.2 | -0.29 | 18.9 | -0.50 |
| GPM | 34.4 | -0.13 | 36.6 | -0.17 | 58.2 | -0.16 | **53.7** | **-0.10** |
| DGP | **36.6** | **-0.12** | **39.2** | **-0.09** | **67.2** | **-0.06** | 48.0 | -0.13 |

Table 3: Comparisons of ACC and BWT after learning all the tasks on the Split-CIFAR100 dataset.

## B.2 ARCHITECTURE DETAILS OF NEURAL NETWORKS

**MLP**: The fully-connected network in Permuted MNIST and Rotated MNIST experiments consists of three linear layers with 256/256/10 hidden units. No bias units are used. The activation function is Relu. Each task has an independent first layer without constraints imposed on its weight update.

**AlexNet**: The modified AlexnetKrizhevsky et al. (2012) in the Split-CIFAR100 experiment consists of three convolutional layers with 32/64/128 kernels of size $(4 \times 4)/(3 \times 3)/(2 \times 2)$, and three fully connected layers with 2048/2048/10 hidden units. No bias units are used. Each convolution layer is followed by a $(2 \times 2)$ average-pooling layer. The dropout rate is 0.2 for the first two convolutional layers and 0.5 for the remaining layers. The activation function is Relu. Each task has an independent first layer and final layer (classifier) without constraints imposed on its weight update.

**ResNet18**: The variant ResNet18Chaudhry et al. (2019b) in the Split-miniImageNet experiment consists of 17 convolutional blocks and one linear layer. The convolutional block comprises a convolutional layer and a batch normalization layer and an Relu activation. The first and last convolutional blocks are followed by a $(2 \times 2)$ average-pooling layer respectively. All convolutional layers use $(1 \times 1)$ zero-padding and kernels of size $(3 \times 3)$. The first convolutional layer has 40 kernels and $(2 \times 2)$ stride, followed by four basic modules, each comprising four convolutional blocks with same number of kernels 40/80/160/320 respectively. The first convolutional layer in each basic modules has $(2 \times 2)$ stride, while the remaining three convolutional layers have $(1 \times 1)$ stride. The skip-connections occur only between basic modules. No bias units are used. In batch normalization layers, tracking mean and variance is used, and the affine parameters are learned in the first task $\mathcal{T}_1$, which are then fixed in subsequent tasks. Each task has an independent first layer and final layer.

## B.3 HYPER-PARAMETER CONFIGURATIONS

### B.3.1 ADVERSARIAL ATTACK ALGORITHM

The norm of attacks used in experiments are $\ell_\infty$. The hyper-parameters in various attack algorithm are provided in Table. 4.

| dataset | Attack method | | |
|---|---|---|---|
| | AutoAttack | PGD | FGSM |
| PMNIST | $\xi = 20/255$ | $\gamma = 2/255, \xi = 40/255$ | $\xi = 25/255$ |
| RMNIST | $\xi = 20/255$ | $\gamma = 2/255, \xi = 40/255$ | $\xi = 25/255$ |
| Split-CIFAR100 | $\xi = 2/255$ | $\gamma = 1/255, \xi = 4/255$ | $\xi = 4/255$ |
| Split-miniImageNet | $\xi = 2/255$ | $\gamma = 1/255, \xi = 4/255$ | $\xi = 2/255$ |

Table 4: Hyper-parameter setup to control the attack strength

The perturbation size used in our experiments are smaller than the typical value in adversarial robustness literature. This adjustment is made because when confronted with such intensity of adversarial attacks, regardless of approaches considered (including baselines and the proposed method), the neural network's robustness on the current task decreases significantly after learning only two or three news tasks. Thus, we slightly reduced the perturbation size. Then, the advantage of the proposed method becomes evident. While most baselines still exhibit a significant decrease after learning only two or three new tasks, the proposed method enables mitigating decrease of robustness after learning a sequence of new tasks.

### B.3.2 CONTINUAL LEARNING ALGORITHM

We outline the fundamental principle for each continual learning algorithm in baselines as follows:

- EWC is a regularization technique that utilizes the Fisher Information matrix to quantify the contribution of model parameters on preserving knowledge of previous tasks;
- SI computes the local impact of model parameters on global loss variations, consolidating crucial synapses by preventing their modification in new tasks;

- A-GEM is a memory-based approach, similar to GEM, which leverages data from episodic memory to adjust the gradient direction of current model update;

- OGD is another gradient projection approach where each base vector constrains the weight updates of the entire model, while GPM employs a layer-wise gradient projection strategy.

We run the methods of SI, EWC, GEM, A-GEM based on the Avalanche Lomonaco et al. (2021), an end-to-end continual learning library. In DGP, $\alpha_1$ and $\alpha_2$ (see $\alpha$ in Eq. 6 of main text) control the number of base vectors added into the pool for stabilizing the final output and sample gradients respectively. $\alpha_3$ is used in reducing the number of base vectors when the pool is full (by performing the SVD and $k$-rank approximation on the matrix consisting of all base vectors in the pool).

| Dataset | Method | Hyperparameter | |
|---|---|---|---|
| | | Learning rate | Others |
| Permuted MNIST | SGD | 0.1 | None |
| | SI | 0.1 | $\lambda = 0.1$ |
| | EWC | 0.1 | $\lambda = 10$ |
| | GEM | 0.05 | patterns_per_exp = 200 |
| | A-GEM | 0.1 | sample_size = 64, patterns_per_exp = 200 |
| | OGD | 0.05 | memory_size = 300 |
| | GPM | 0.05 | memory_size = 300, $\alpha1 = [0.95, 0.99, 0.99]$ |
| | DGP | 0.05 | memory_size = 300, $\alpha1 = [0.95, 0.99, 0.99]$, $\alpha2 = 0.999, \alpha3 = 0.996$ |
| Rotated MNIST | SGD | 0.1 | None |
| | SI | 0.1 | $\lambda = 0.1$ |
| | EWC | 0.1 | $\lambda = 10$ |
| | GEM | 0.05 | patterns_per_exp = 200 |
| | A-GEM | 0.1 | sample_size = 64, patterns_per_exp = 200 |
| | OGD | 0.05 | memory_size = 300 |
| | GPM | 0.05 | memory_size = 300, $\alpha1 = [0.95, 0.99, 0.99]$ |
| | DGP | 0.05 | memory_size = 300, $\alpha1 = [0.95, 0.99, 0.99]$, $\alpha2 = 0.999, \alpha3 = 0.996$ |
| Split-CIFAR100 | SGD | 0.05 | None |
| | SI | | $\lambda = 0.1$ |
| | EWC | | $\lambda = 10$ |
| | GEM | | patterns_per_exp = 200 |
| | A-GEM | | sample_size = 64, patterns_per_exp = 200 |
| | OGD | | memory_size = 300 |
| | GPM | | memory_size = 100, $\alpha1 = 0.97 + 0.003*$task_id |
| | DGP | | memory_size = 100, $\alpha1 = 0.97 + 0.003*$task_id, $\alpha2 = 0.996, \alpha3 = 0.99$ |
| Split-miniImageNet | SGD | 0.1 | None |
| | SI | | $\lambda = 0.1$ |
| | EWC | | $\lambda = 10$ |
| | GEM | | patterns_per_exp = 200 |
| | A-GEM | | sample_size = 64, patterns_per_exp = 200 |
| | OGD | | memory_size = 100 |
| | GPM | | memory_size = 100, $\alpha1 = 0.985 + 0.003*$task_id |
| | DGP | | memory_size = 100, $\alpha1 = 0.96$, $\alpha2 = 0.996, \alpha3 = 0.996$ |

Table 5: Hyper-parameter setup in our approach and other CL algorithms in baselines.

## B.4 INCOMPATIBILITY BETWEEN EXISTING CONTINUAL LEARNING AND DEFENSE ALGORITHMS

Distillation is a well-known defense method Carlini & Wagner (2017) that involves training two models - a teacher model is trained using one-hot ground truth labels and a student model is trained using the softmax probability outputs of the teacher model. The result of combinations of Distillation and existing continual learning algorithms are presented in Fig. 8. There is a notable trend in Fig.8d: the blue line (representing the performance of Distill+GPM on original samples) exhibits a more rapid decline compared to the corresponding blue line in the fourth subplot of the first row in Fig.3 (representing IGR+GPM), as well as the pink line in Fig.4d (representing AT+GPM). Additionally, the purple and blue lines in Fig.8d (representing Distill+GEM and Distill+GPM) closely align with the green line in Fig.8d (representing Distill+SGD). These observations suggest again incorporating the defense algorithms, such as Distillation, into the training procedure compromise the efficacy of these continual learning methods.

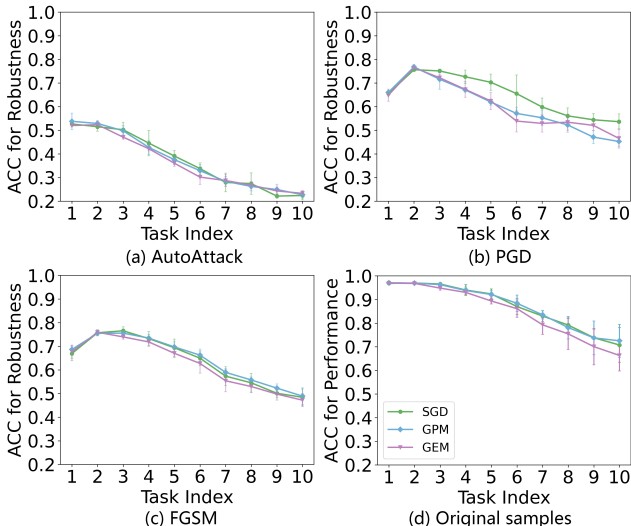

Figure 8: As Fig. 3, but for defense algorithm Distillation on PMNIST dataset. Here, we combine Distillation with continual learning algorithms GEM and GPM, which have shown superior ACC compared to other baselines in Fig. 3.

