# OpenReview forum: "Maintaining Adversarial Robustness in  Continuous Learning"
_ICLR.cc/2025/Conference — Submitted to ICLR 2025_

### Official Review · Reviewer_5kyL · 2024-10-31

**Soundness:** 2
**Presentation:** 2
**Contribution:** 2
**Rating:** 3
**Confidence:** 4

**Summary:**

This studies the adversarial robustness of image classifiers in the setting of continual learning. The aim is to maintain the adversarial robustness across a series of tasks as the model continues to learn on the new tasks. It proposes a method to stabilize the sample gradients smoothed by defense algorithms such as IGR and AT on previous tasks.

**Strengths:**

1. Adversarial robustness in continual learning is a meaningful direction to explore and it has been less explored before.
2. Based on the given results, the proposed method seem to be effective. However, the reliability of conducted evaluation may be problematic as analyzed below.

**Weaknesses:**

1. The presentation is hard to follow. For example, the section 3 starts with a subsection named "Constraint on Weight Updates" without any explanation or context like what this subsection is about, why here.
2. Several important discussions are missing. First, the authors claim as a contribution introducing the problem of robust continual learning, however, this setting is not formally defined in the paper. Second, the proposed method hypothesizes that "if we can stabilize the sample gradients smoothed by defense algorithms such as IGR and AT on previous tasks, the adversarial robustness of the neural network will hold even after its weights update for learning a sequence of new tasks." However, no discussion is provided to explain why.
3. In the experiments, an independent first layer was assigned to each task. Is this necessary for the proposed method? If so, this complicates the method and restricts its application scenarios.
4.  The main result is based on the defense algorithm IGR which, however, is dated, so the result may not reflect the state-of-the-art. Besides, regularization-based defense like IGR is less effective than adversarial training defense according to the results on RobustBench. Therefore, I feel reporting main results on IGR is less meaningful.
5. The reported results seem to be promblematic. According to Tab. 4 in appendix, the perturbation budget is 2/255 for AutoAttack and 4/255 for PGD and FGSM on Split-CIFAR100 and Split-miniImageNet. Given the great difference in the budget, the robust accuracy evaluated by them should also exhibit obvious gap. However, as shown in 3rd row in Fig.3, such gap is not observed. Moreover, PGD is a much stronger than FGSM so the robust accuracy evaluated by PGD should be lower than that of FGSM. However, again in the same row and figure, PGD accuracy is higher than FGSM accuracy. Overall, I feel that the aversarial evaluation in this work is not reliable.
6. The discussion of related works is insufficient, covering only one related work and two surveys. It misses some related works, e.g., alternative formulation of adversarial continual learning [1], zero-shot and few-shot adversarial robustness [2, 3].

[1] Zhou, Yuhang, and Zhongyun Hua. "Defense without Forgetting: Continual Adversarial Defense with Anisotropic & Isotropic Pseudo Replay." Proceedings of the IEEE/CVF Conference on Computer Vision and Pattern Recognition. 2024.

[2] Li, Lin, et al. "One prompt word is enough to boost adversarial robustness for pre-trained vision-language models." Proceedings of the IEEE/CVF Conference on Computer Vision and Pattern Recognition. 2024.

[3] Mao, Chengzhi, et al. "Understanding Zero-shot Adversarial Robustness for Large-Scale Models." The Eleventh International Conference on Learning Representations. 2023.

**Questions:**

1. at Line 309, AT is cited to the publication "Adversarial examples in the physical world". To my best knowledge, that publication does not give an algorithm of adversarial training. It is also not a common citation of AT.

---

### Official Review · Reviewer_L1v2 · 2024-11-03

**Soundness:** 2
**Presentation:** 2
**Contribution:** 2
**Rating:** 3
**Confidence:** 4

**Summary:**

This paper aims to address the issue of maintaining adversarial robustness in continual learning through the proposed gradient projection technique. The authors demonstrate the effectiveness of their technique combined with several existing defense algorithms across multiple benchmarks. However, despite providing valuable insights and positive initial results, the paper has noticeable deficiencies and limitations in some critical areas.

**Strengths:**

1. The proposed Double Gradient Projection (DGP) technique is theoretically innovative, stabilizing sample gradients from previous tasks by orthogonally constraining the direction of weight updates.
2. Experiments conducted on four different datasets validate the method’s ability to maintain adversarial robustness in continual learning tasks, especially under strong adversarial attacks.
3.The paper builds on theories of gradient smoothing and gradient projection to provide a plausible explanation for maintaining adversarial robustness during continual learning.

**Weaknesses:**

1. The paper claims to introduce the problem of robust continual learning and perform robust continual learning without access to historical data. However, the gradient memory used in the proposed method actually amounts to indirect access to previous data, which contradicts the initial claim to some extent.
2. Although the authors claims that the proposed technique can enhance robustness when combined with existing defense methods, the paper lacks experimental results after integration with the latest defense strategies, weakening the persuasiveness of the method.
3. The literature review on related work in the paper is very limited, mentioning only a few relevant studies. Without extensive literature comparisons, it is difficult to demonstrate the effectiveness and innovation of the proposed technique.
4. The computational cost introduced by performing singular value decomposition on layer gradients has not been adequately assessed. Given the importance of computational efficiency in practical applications, neglecting this aspect could impact the method’s practicality.
5. The abstract and several sections of the manuscript contain grammatical errors.

**Questions:**

See the weaknesses above.

---

### Official Review · Reviewer_zjL9 · 2024-11-04

**Soundness:** 2
**Presentation:** 1
**Contribution:** 1
**Rating:** 1
**Confidence:** 4

**Summary:**

This paper studies the problem of improving adversarial robustness in the context of continual learning. The method of Double Gradient Projection (DGP) is proposed to mitigate the rapid degradation of adversarial robustness during continual learning. The authors conduct experiments using the variants of dataset MNIST, CIFAR, and MiniImageNet, to show the proposed method is better than the naive combination of adversarial defense and continual learning methods.

**Strengths:**

1. Investigated a relatively new problem.
2. Studying adversarial robustness from the gradient perspective is pertinent.

**Weaknesses:**

1. The investigated task seems like a simple combination of two well-studied problems, adversarial learning and continual learning. I did not see any practical necessity in studying this task in your paper.
2. The experiments need to be improved. Only evaluated on several small datasets, with very old networks like AlexNet and ResNet18.
3. Only considered defense algorithms such as IGR and AT. It is not clear how more recent defense methods perform under the continual learning context.
4. The proposed method does not introduce any new insights into both adversarial learning and continual learning.

**Questions:**

1. Why do you introduce this task? Can you name a real case where this task is necessary?
2. What is the biggest challenge for maintaining adversarial robustness in continual tasks?
3. What motivates your proposed method? It's confusing why you suddenly put forward this Double Gradient Projection method to handle your task.

---

### Official Review · Reviewer_o3jn · 2024-11-04

**Soundness:** 3
**Presentation:** 3
**Contribution:** 3
**Rating:** 6
**Confidence:** 3

**Summary:**

The paper introduces Double Gradient Projection (DGP), a method designed to maintain adversarial robustness in continuous learning settings where neural networks often suffer from catastrophic forgetting. By stabilizing sample gradients through orthogonal projections during weight updates, DGP effectively mitigates the loss of robustness when learning new tasks, as demonstrated in experiments on benchmarks like Split-CIFAR100 and Split-miniImageNet.

**Strengths:**

1. The author proposed a novel gradient projection technique to maintain the adversarial robustness in a continuous learning setting.
2. Experimental results on several benchmarks demonstrate the superiority of DGP over existing approaches.
3. It would be useful in the scenario that the data of previous tasks can not be revisited after training.

**Weaknesses:**

1. I am concerned the computational complexity increases due to the need for singular value decomposition (SVD), especially when using larger neural networks, which are supposed to have more capacity and are suitable for learning multi-tasks. Given that the perturbation size is also smaller and larger storing space of bases, I may doubt the practice usage of the proposed method.

2. The conducted experiments are on small datasets and small-scale networks.

3. A brief introduction to the dataset used is necessary, which can help the reader understand the scenario and tasks.

4. Some literature is missing like [1][2].

5. What kind of Adversarial Training is applied? The literature on adversarial training is also quite limited.

[1] Adversarial Continual Learning, ECCV.

[2] Retrospective Adversarial Replay for Continual Learning, NeurIPS.

**Questions:**

1. It would be good to see the distribution of k values for different tasks, can it represent the difficulty of transferability from the previous task? Also, does the sequence of tasks index matter?
2. Can DGP also be combined with EWC/IGR like GEM+IGR shown in Fig 3? This could further provide more understanding of where the benefits come from.

---

### Meta-Review · Area_Chair_2wPx · 2024-12-19

**Metareview:**

This work investigates maintaining adversarial robustness within the framework of continuous learning. During the rebuttal phase, reviewers raised several significant concerns. Firstly, there are practical applicability issues related to the proposed approach's high computational complexity and increased storage requirements. Additionally, questions were raised about the experimental settings and results, highlighting that the experiments were conducted on small-scale datasets and networks, the defense algorithms employed are not up-to-date, and there are doubts regarding the validity of the experimental performance. The literature review was also considered to be insufficient, lacking comprehensive coverage of relevant studies. Furthermore, some reviewers expressed concerns about the technical contributions and novelty of the work, suggesting that the proposed method appears to be a combination of adversarial training and continuous learning, offering limited new insights to the community. During the rebuttal phase, the authors did not provide any responses to these concerns. Consequently, the paper cannot be accepted unless these issues are adequately addressed.

**Additional Comments On Reviewer Discussion:**

The authors did not provide responses during the rebuttal phase.

---

### Decision · Program_Chairs · 2025-01-22

Reject